# Surface Water Salinity Evaluation and Identification for Using Remote Sensing Data and Machine Learning Approach

**Raisa Borovskaya** [1], **Denis Krivoguz** [1], **Sergei Chernyi** [2,3,4,*], **Efim Kozhurin** [1], **Victoria Khorosheltseva** [1,5] and **Elena Zinchenko** [2,3]

1. Russian Federal Research Institute of Fisheries and Oceanography (FSBSI "VNIRO"), Azov-Black Sea Branch of the FSBSI "VNIRO" ("AzNIIRKH"), 344000 Rostov-on-Don, Russia; raisaborovskaya@vniro.ru (R.B.); krivoguzdenis@gmail.com (D.K.); kozhurin_e_a@azniirkh.ru (E.K.); horosheltseva_v_n@azniirkh.ru (V.K.)
2. Department of Ship's Electrical Equipment and Automatization, Kerch State Maritime Technological University, 298309 Kerch, Russia; eltel85@bk.ru
3. Department of Cyber-Physical Systems, St. Petersburg State Marine Technical University, 190121 Saint-Petersburg, Russia
4. Department of Complex Information Security, Admiral Makarov State University of Maritime and Inland Shipping, 198035 Saint-Petersburg, Russia
5. Department of Zoology, Southern Federal University, 344000 Rostov-on-Don, Russia
* Correspondence: chernysg@gumrf.ru or sergiiblack@gmail.com

**Abstract:** Knowledge of the spatio-temporal distribution of salinity provides valuable information for understanding different processes between biota and environment, especially in hypersaline lakes. Remote sensing techniques have been used for monitoring different components of the environment. Currently, one of the biggest challenges is the spatio-temporal monitoring of the salinity level in water bodies. Due to some limitations, such as the inability to be located there permanently, it is difficult to obtain these data directly. In this study, machine learning techniques were used to evaluate the salinity level in hypersaline East Sivash Bay. In total, 93 in situ data samples and 6 Sentinel-2 datasets were used, according to field measurements. Using linear regression, random forest and AdaBoost models, eight water salinity evaluation models were built (six with simple, one with random forest and one with AdaBoost). The accuracy of the best-fitted simple linear regression model was 0.8797; for random forest, it was equal, at 0.808, and for AdaBoost, it was −0.72. Furthermore, it was found that with an increase in salinity, the absorbing light shifts from the ultraviolet part of the spectrum to the infrared and short-wave infrared parts, which makes it possible to produce continuous monitoring of hypersaline water bodies using remote sensing data.

**Keywords:** correlation analysis; GIS; linear regression; remote sensing; maritime; water salinity

## 1. Introduction

Water salinity, along with its temperature, is the most important factor in determining the level of density in seawater and, consequently, the movement of water masses in the world's oceans. The most important factor affecting the salinity levels in the surface layer of any water body is the level of precipitation and evaporation [1]. It should also be noted that the salinity level and its dynamics can indirectly describe the states and structures of aquatic ecosystems.

Salinity belongs to the group of environmental factors that significantly affect the formation of a particular type of coastal marine ecosystem [2]. A number of studies shows that in some cases, triggered by anthropogenic impact, such as a decrease in river flow as a result of watercourses regulation, as well as natural causes, salinity can be a limitation factor for marine organisms.

The average salinity of sea water is about 35 g/L, but in many parts of the oceans, this value differs significantly. Since marine flora and fauna depend on the level of water salinity, salinity largely determines the ecological type of water objects.

Changes in the level of salinity in a water body can be a consequence of weather events, such as droughts or floods, or it can be caused by an increase in anthropogenic pressure on the environment, the excessive regulation of rivers, an increase in the level of wastewater discharge into water bodies, etc. [3].

Many marine organisms are highly dependent on salinity level changes. The main causes of this are the peculiarities of the osmosis process, which is penetration into the living cells of the body and vice versa, depending on the concentration of substances dissolved in water, until equilibrium is reached [4].

The increase in salinity levels in Sivash Bay after 2014 led to significant changes not only in the composition of the benthos community, but also in the entire ecosystem. Before this, such a high species diversity in hypersaline water bodies had not been observed. Thus, in Sivash Bay we can now see the phenomenon of hysteresis, when different types of biodiversity can be observed at the same salinity level, i.e., if salinity rises, the diversity will be higher than when it falls [5].

Determining salinity levels is the most important step in planning fishing and fish farming in a particular body of water [4].

This indicator is of particular importance for aquaculture. The selection of the most suitable fish species for breeding is based on the biological features of the species. The main limiting factor in seas, bays and estuarine zones is salinity. In this case, the choice of aquaculture object is made after determining the main hydrological indicators, one of which is salinity. For example, some fish species (pangasius, carp) have a low range of tolerance to salinity (up to 5 g/L), while other species (Atlantic salmon, tilapia, rainbow trout) show high growth rates in waters with salinity of up to 20 g/L [6,7].

Determining the salinity level in a water body allows the prediction of the values of other hydrochemical indicators. For example, salinity affects the level of dissolved oxygen in water. Dissolved oxygen in water decreases with increasing salinity. The solubility of oxygen in seawater is about 20% less than in fresh water at the same temperature. If hypersaline reservoirs are used, it is preferable to choose fish species that are tolerant to low oxygen levels in water.

### 1.1. Water Salinity and Its Absorption Properties

The absorption of the world's oceans is associated with the quantum-mechanical interactions of electromagnetic radiation propagating with the atoms and molecules of matter in the environment. As a result of these interactions, part of the energy at certain frequencies is transferred to atoms and molecules, which manifests itself in the formation of absorption bands in the light spectrum. The absorption index $k$ is determined through the imaginary part of the complex refractive index $n = m - ik$:

$$k(\lambda) = (4 * \pi \lambda)k(\lambda) \tag{1}$$

Water has a strong dissonant ability, so the molecules of inorganic salts break down into ions when they are dissolved in water. The ions of inorganic salts have an electronic absorption spectrum located in their ultraviolet parts. As a consequence, an increase in the concentration of salts noticeably affects the increase in the absorption in the ultraviolet part. The main reason for this effect is the presence of bromide compounds and nitrates. In their visible and infrared parts, inorganic salts have a noticeable effect on the absorption index.

According to a study by C.D. Mobley's [8], in which he studied the absorption of water in a wavelength range from 0.01 cm (X-rays) to 1 m (radio-waves), in the context of ocean optics, only the range from near-ultraviolet (near-UV) to near-infrared (near-IR) was shown to be of scientific interest. R.C. Smith and K.S. Baker defined this range more precisely. According to their study [9], it ranges from 200 nm to 800 nm. They also note that the absorption is slightly dependent on the level of salinity, at least in the red and infrared parts of the spectrum.

### 1.2. Remote Sensing of Different Environmental Variables

In recent years, several studies have been carried out concerning the acquisition of environmental data using remote sensing. An important approach is to obtain the Earth's surface temperature data. Great progress in this direction was made by V. Solanky et al. [10]. In their research, they tried to obtain the Earth's surface temperature data using MODIS and Landsat-8 satellite imagery for the Bhopal region.

From the oceanographical point of view, it is also important to obtain the chlorophyll-a concentration in the surface layer, which makes it possible to solve many problems in ocean biology, fisheries, etc. The research performed by M.A. Matus-Hernandez et al. [11] using linear and multiple regression and generalized additive models (GAM) for the predictive modeling of chlorophyll concentrations using Landsat data showed good accuracy and can be used in future studies.

With regard to water salinity, the study by D. Sun et al. [12] on using remote sensing data to determine sea surface salinity in the Southern Yellow Sea and F. Wang and Y.J. Xu's study [13] on using remote sensing data to determine estuarine water salinity showed that their models have great potential for mapping sea surface salinity.

It should also be noted that here are no significant studies and that the physico-chemical features of these water bodies make it impossible to extrapolate the experience of evaluating the salinity level using remote sensing data.

### 1.3. Deep Learning Approach for Evaluating Water Salinity Using Remote Sensing Data

One of the modern approaches to the evaluation of environmental properties through remote sensing is to use deep learning. The basis of deep learning is neural networks with different architectures, depending on the task. The basic scheme (Figure 1) for evaluating water salinity, or any other environmental variable form remote sensing data using deep learning, is similar to other machine learning model, such as linear regression, support vector machine, etc. The first step in this approach is to collect field data from water objects in different locations and extract remote sensing data from the same locations from.

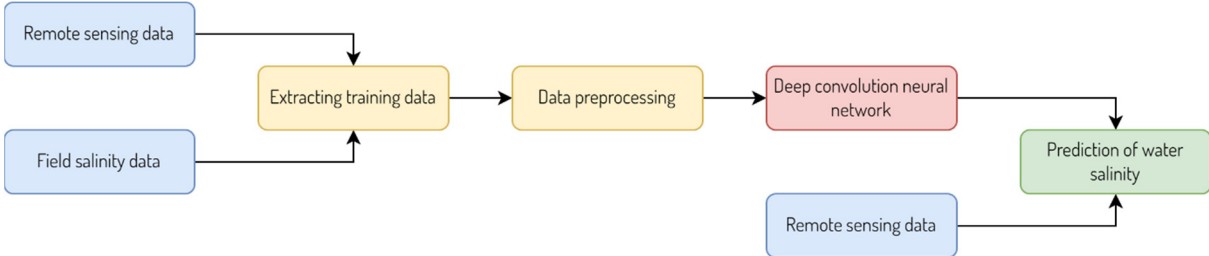

**Figure 1.** Scheme for evaluating water salinity from remote sensing data using deep convolution neural network.

The next step is to preprocess where we perform data normalization and standardization while excluding missing values. The most common methods for data normalization are: Minimax—linear data transformation in the range of 0–1, where the minimum and maximum scalable values correspond to 0 and 1; and Z-scaling, which is based on the mean and standard deviation and involves dividing the difference between the variable and the means by standard deviation and decimal scaling, which is performed by removing the decimal separator of the variable value. In practice, Minimax and Z-scaling have similar areas of applicability and are often interchangeable. However, when calculating the distances between points or vectors in most cases, Z-scaling is used, while minimax is useful for visualization.

The use of deep convolution neural network (CNN) for evaluating water salinity from remote sensing data (Figure 2) consists of input data from in situ salinity data and spectral values of different channels of satellite images. Next, the input data procced to convolution and pooling layers to train the CNN and obtain the predicted salinity values.

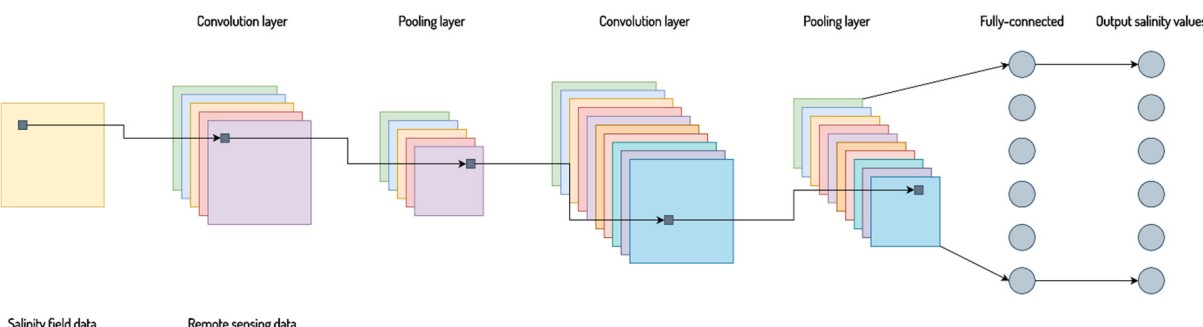

**Figure 2.** Schematic visualization of deep convolution neural network architecture for evaluating water salinity.

Therefore, our goal was to investigate the possibility of evaluating salinity using remote sensing data for hypersaline waterbodies, taking into account research experience with similar objects.

## 2. Materials and Methods

### 2.1. Research Area

The Sivash is a vast shallow bay of the Sea of Azov with a fairly indented coastline and a large number of peninsulas, capes and bays [14]. The Sivash is connected with the Sea of Azov by the shallow and narrow Genichesk Strait, which is located near the northern part of the Arabat Spit (Figure 3). The length of the strait is 5 km, the width is 80–120 m, and the depth is about 2.0–3.5 m. The area of the Sivash is about 2500 km$^2$ [15].

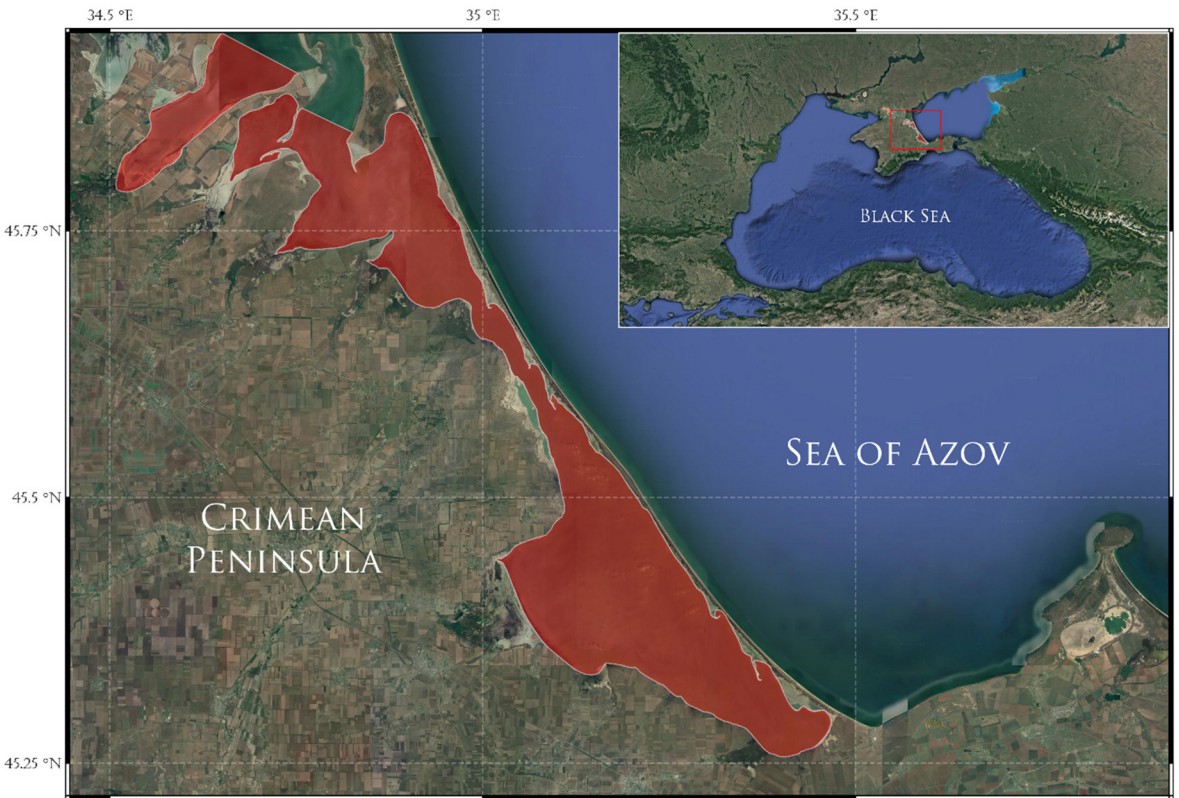

**Figure 3.** Schematic map of the research area.

The Sivash is separated from the open sea by the Arabat Spit. The Sivash is divided into two parts by the Chontarsky peninsula: Western and Eastern. These parts are connected by the Chontarsky Strait (about 1 km long, 200–300 m wide and 0.5–1 m deep). The Western

part of the Sivash ranges from the Chongarsky peninsula to the Perekop Isthmus [16]. The total length of the Western part is about 80 km, and the width ranges from 5 to 20 km. In some cases, the part from the Chongarsky Peninsula to Cape Kurgan is called the Middle Sivash, and the part between Cape Kurgan and the Perekop Isthmus is sometimes called Western Sivash.

The Eastern part of the Sivash is located along the Arabat Spit from the Genichesk to Rybatsloye village. The total length of the Eastern Sivash is about 120 km, while its width is 2–35 km. The widest area is located near Rogozinsky Bay. The Eastern part is deeper than the Western one [17].

Sivash Bay is a source of many important industrial resources, which include bromine and its compounds, salt, titanium dioxide, phosphoric fertilizers, iron oxide pigment, sulfuric acid and copper sulfate.

### 2.2. Field Data and Research

An annual study of the state of the aquatic environment in the Sivash has been performed by the Azov-Black Sea branch of the VNIRO ("AzNIIRKH") since 2017 in the spring (May), summer (July) and autumn (October). The data used are based on the results of field research and monitoring for the period from 2018 to 2019 in the eastern part of Sivash Bay at 14 stations, the schematic locations of which are shown in Figure 4. The total number of samples taken over 2 years was 93. All salinity samples were collected in separate bottles and analyzed in a laboratory using a 601-MK-III salinometer.

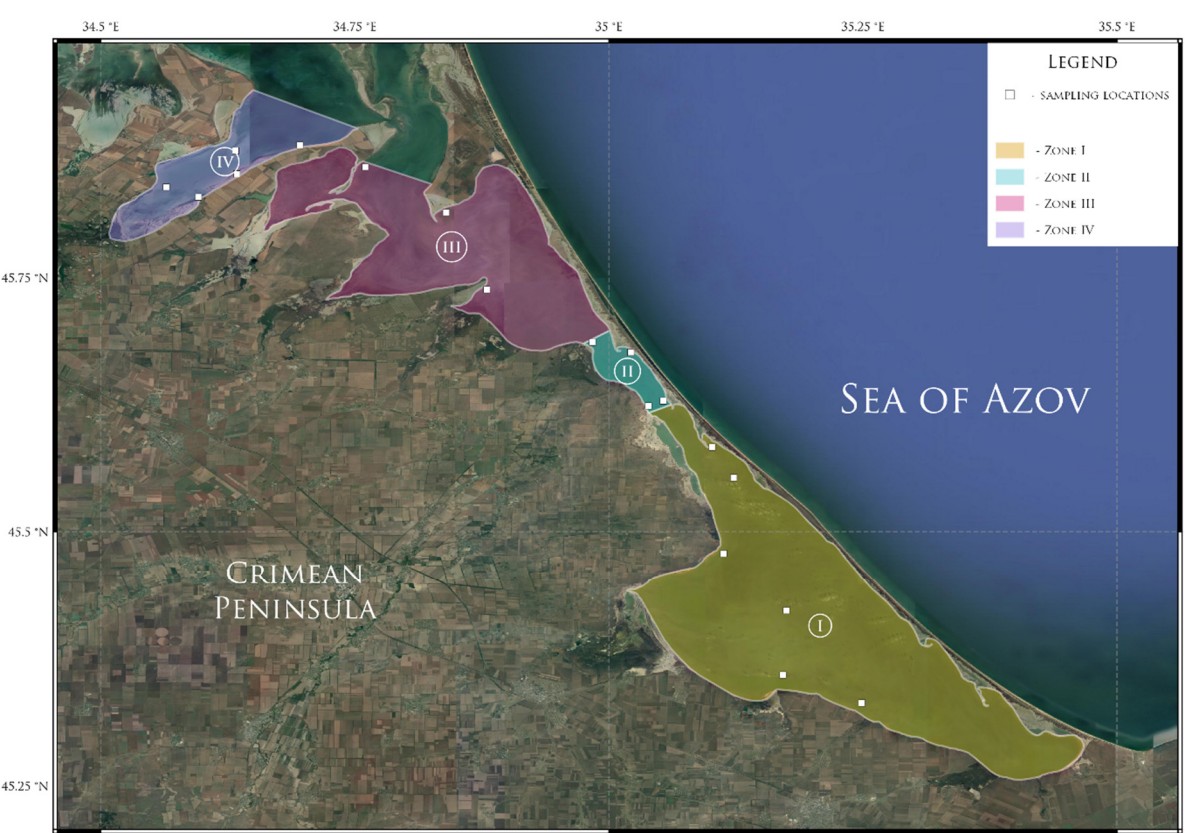

**Figure 4.** Schematic map of monitoring field stations on East Sivash Bay.

### 2.3. Remote Sensing Data

In this study, the authors used Sentinel-2 satellite data, downloaded from ESA Sci-hub data portal. Sentinel-2 is the European Space Agency's family of Earth remote sensing satellites, created as part of the "Copernicus" global environmental and safety monitoring project.

The data received by the Sentinel-2 satellite consists of 13 spectral channels: 4 channels with a spatial resolution of 10 m, 6 channels with a spatial resolution of 20 m and 3 channels with a spatial resolution of 60 m [18].

The orbit, with an altitude of about 285 km, as well as the presence of two satellites on the orbit, makes it possible to carry out continuous surveying of the Earth's surface with a frequency of about 5 days in the equatorial regions and about 2 or 3 days in mid-latitude areas.

In this study, we used 6 Sentinel-2 datasets, sensed around the dates of sampling of in situ data, presented in Table 1.

**Table 1.** Satellite images used in this study.

| Name | Date of Sensing |
| --- | --- |
| S2A MSIL1C 20181006T082811 N0206 R021 T36TXR 20181006T094943 | 6 October 2018 |
| S2A MSIL2A 20190606T083601 N0212 R064 T36TXR 20190606T112641 | 6 June 2019 |
| S2A MSIL2A 20190706T083611 N0212 R064 T36TXR 20190706T112755 | 6 July 2019 |
| S2A MSIL1C 20190723T082611 N0208 R021 T36TXR 20190723T103117 | 23 July 2019 |
| S2A MSIL1C 20180522T083601 N0206 R064 T36TXR 20180522T105656 | 22 May 2018 |
| S2A MSIL1C 20190519T082609 N0207 R021 T36TXR 20190519T112607 | 19 May 2019 |

### 2.4. Satellite Data Preprocessing

Preprocessing of satellite images is an important stage of research using remote sensing data.

Our preprocessing pipeline included conversion of brightness values to the top of the atmosphere radiance and atmospheric correction. We used a "sen2cor" processor for Sentinel-2 level 2A and level 1C data for atmospheric, terrain and cirrus correction.

All satellite data were chosen according to the time at which the water samples were collected in Sivash Bay to decrease the impact of any changes in environmental conditions.

### 2.5. Validation of Prediction Results

The quality of the models used in this work was checked with $R^2$ and adjusted $R^2$. The coefficient of determination ($R^2$) shows how much the conditional variance of the model differs from the real values [19]. If this coefficient is close to 1, then the conditional value of the model's variance is quite small and most likely the model describes the data well enough [20]. If $R^2$ is significantly less than 1, then, with a high level of probability, the model does not reflect the real situation [21].

However, despite its undeniable advantages, $R^2$ also has a serious disadvantage—with an increase in the amount of predictors, it can only increase [22]. Therefore, it may seem that a model with more predictors is better than a model with fewer, even if all new predictors do not affect the variable.

Therefore, using $R^2$ is acceptable only for describing the results of simple linear regression modeling [23]. When using multiple regression, it is optimal to use the adjusted $R^2$ ($R_{adj}^2$). The reason lies in the fact that if new independent variables create a large impact on the model's quality, the value of this factor increases and, if not, it decreases. $R^2$ and $R_{adj}^2$ are defined as:

$$R^2 = 1 - \frac{\sum (\hat{y}_i - y_i)}{\sum (y_i - \overline{y}_i)^2} \tag{2}$$

and

$$R_{adj}^2 = 1 - \left(1 - R^2\right) \times \frac{k-1}{k-n-1} \tag{3}$$

where $y_i$ is the real value of $y$ in each observation, $\hat{y}$ is the value predicted by the model, $\overline{y}$ is an average overall value of $y$, $n$ is the number of samples in the dataset and $k$ is the number of independent variables in the model.

*2.6. Factor Selection*

To select factors that are suitable for building linear models, we used correlation analysis. Its main goal is to investigate connections between different variables to reflect their relations. Generally, correlation analysis is a statistical method that helps to define any dependency between variables and its strength using correlation coefficients. Its results can be interpreted using the Chaddock scale (Table 2).

**Table 2.** Chaddock scale for interpretation of correlation analysis results.

| Absolute Value of Correlation, $|R|$ | Interpretation |
| --- | --- |
| 0.00–0.30 | Negligible correlation |
| 0.30–0.50 | Weak correlation |
| 0.50–0.70 | Moderate correlation |
| 0.70–0.90 | Strong correlation |
| 0.90–1.00 | Very strong correlation |

As the correlation measure, we used Pearson's correlation test, which is calculated as:

$$r = \frac{\sum (x - m_x)(y - m_y)}{\sqrt{\sum (x - m_x)^2 \sum (y - m_y)^2}} \tag{4}$$

where $x$ and $y$ are variables, while $m_x$ and $m_y$ are the means of $x$ and $y$, respectively.

To compute correlation coefficients for our variables, we used "cor.test" function in *R* package "stats" [9]. In case of simple linear regression, we chose the five best correlated with salinity level variables.

## 3. Results and Discussion

For the analysis, we used all of the 13-th spectral bands from the Sentinel-2 scenes (Table 1) and several commonly used spectral indices: the Soil-Adjusted Vegetation Index (SAVI), the Normalized Difference Vegetation Index (NDVI), the Vegetation Soil Salinity Index (VSSI) and the Normalized Difference Snow Index (NDSI). The main reason why we used different spectral indices is the fact that they can determine more complex relationships between salinity values and the reflectance from remote sensing data. We hypothesized that the most indicative relationship between remote sensing data and salinity values would have a very high level of correlation ($|r|$ = 0.9–1.0), according to the Chaddock scale (Table 2).

The results of this analysis (Table 3) show that the most strongly correlated predictors with the salinity values are the B11, B07, B8A, B06, B12, B08 channels of the Sentinel-2 satellite. Moreover, we can clearly see the tendency of salinity in hypersaline water bodies to be related to the absorption capacity of water with a reflection from 730 to 2280 nm. The highest absorption level occurs in the range from 785 to 1655 nm, which corresponds to the near- and short-wave infrared parts of the light spectrum. This corresponds with the hypothesis of C.D. Mobley [8], according to which, with an increase in the salinity level of water object, the absorption range will shift to the right part of the light spectrum and the reflection wavelength will increase.

**Table 3.** Results of correlation analysis between potential predictors and in situ salinity samples.

| Predictor | Correlation Value | Predictor | Correlation Value |
|---|---|---|---|
| SAVI (L = 0.75) | 0.92455610 | NDVI | 0.66010546 |
| VSSI | −0.62641882 | SI | 0.77483596 |
| NDSI | −0.66010546 | BI | 0.63046845 |
| B11 | 0.93790118 | B8A | 0.93283050 |
| B07 | 0.91156181 | B06 | 0.91320940 |
| B05 | 0.86980514 | B12 | 0.91387448 |
| TCI 1 | 0.53698365 | TCI 2 | 0.78227702 |
| TCI 3 | 0.21116625 | B08 | 0.93258507 |
| B04 | 0.53790465 | B03 | 0.77959658 |
| B02 | 0.21628457 | | |

To evaluate the salinity values using the remote sensing data, we used simple linear regression, AdaBoost and random forest models.

The main idea behind using linear regression model was to find the dependence level of two or more predictors $x$ on the dependent variable $y$. This dependence level is expressed using the multiple correlation coefficient $r$, which is equal to the square root of the coefficient of determination, which lies in the range from 0 to 1. The model parameters were adjusted for the best level of the model's approximation. In this case, the indicator of the prediction quality is the determination coefficient ($R^2$), i.e., the sum of the squares of the difference between the values of the model and the dependent variable.

As a result, we found regression equations for each of the analyzed predictors. We can see that the highest accuracy value and the highest evaluating potential belonged to the B11 band (Table 4). The results of the correspondence between the predicted salinity values of the water salinity index (WSI) and the field observations are presented in Figure 5.

**Table 4.** Results of the simple linear regression modeling of salinity values.

| Predictor | Regression Equation | $R^2$ |
|---|---|---|
| SAVI | $WSI = 0.913 * SAVI + 3.9042$ | 0.8548 |
| B06 | $WSI = 0.1359 * B06 + 6.191$ | 0.834 |
| B07 | $WSI = 0.1295 * B07 + 8.2156$ | 0.8309 |
| B08 | $WSI = 0.1584 * B08 + 4.4265$ | 0.8697 |
| B8A | $WSI = 0.1597 * B8A + 8.8705$ | 0.8797 |
| B11 | $WSI = 0.1946 * B11 + 14.66$ | 0.8797 |
| B12 | $WSI = 0.1934 * B12 + 18.416$ | 0.8352 |

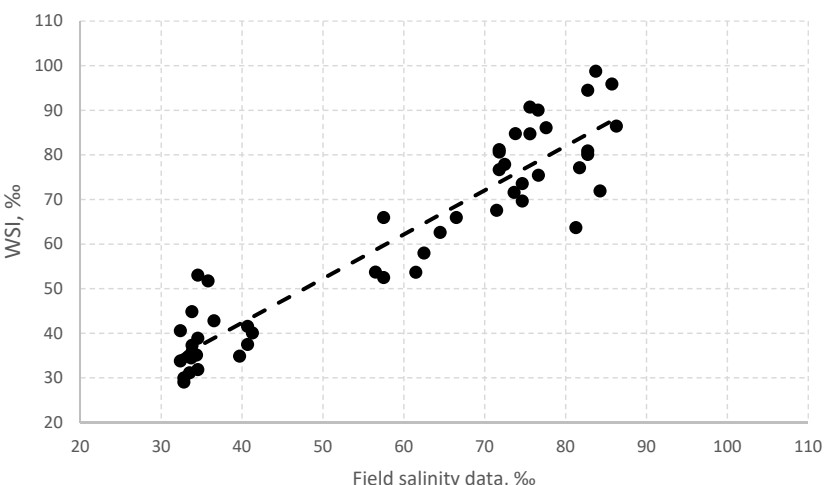

**Figure 5.** Correspondence between salinity values of the water salinity index (WSI) predicted with linear regression using B11 spectral channel and field observations.

The AdaBoost algorithm is a meta-algorithm that, in the learning phase, builds ensembles of weak learning algorithms to increase prediction quality [24,25]. In this case, every subsequent classifier is built based on objects that have not been well classified by previous classifiers. In our case, we predicted salinity values using the AdaBoost algorithm based on the following equation [26,27]:

$$F(x) = sign\left(\sum_{m=1}^{M} \theta_m f_m(x)\right),$$

where $f_m$ is $m$-th weak classifier and $m$ is the weight of the classifier. The accuracy of the prediction of water salinity with the AdaBoost algorithm ($R^2$) was 0.72 (Figure 6).

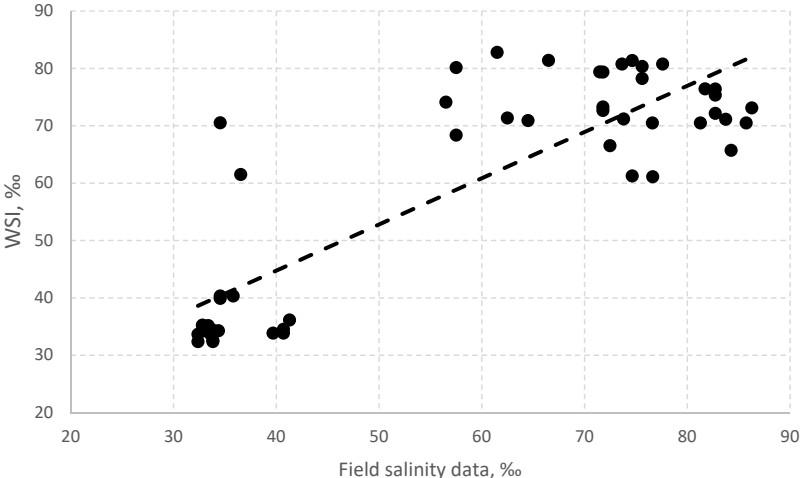

**Figure 6.** Correspondence between salinity values of the water salinity index (WSI) predicted with AdaBoost and field observations.

Random forest is an algorithm that uses a group of decision trees for prediction. In the learning phase, each tree learns of a random sample from dataset [28,29]. The choice of sample is based on bootstrapping [30]. Although each tree vary greatly depending on the

training dataset, learning on a different sample helps to decrease the total variability of the forest. The final results of the random forest model were calculated as [31–33]:

$$a(x) = \frac{1}{N} \sum_{i=1}^{N} b_i(x),$$

where $N$ is the number of trees, $i$ is the trees enumerator, $b$ is the decision tree and $x$ is the generated dataset. A comparison of the predicted salinity data using random forest model and field measurements presented in Figure 7. The total accuracy ($R^2$) was 0.808.

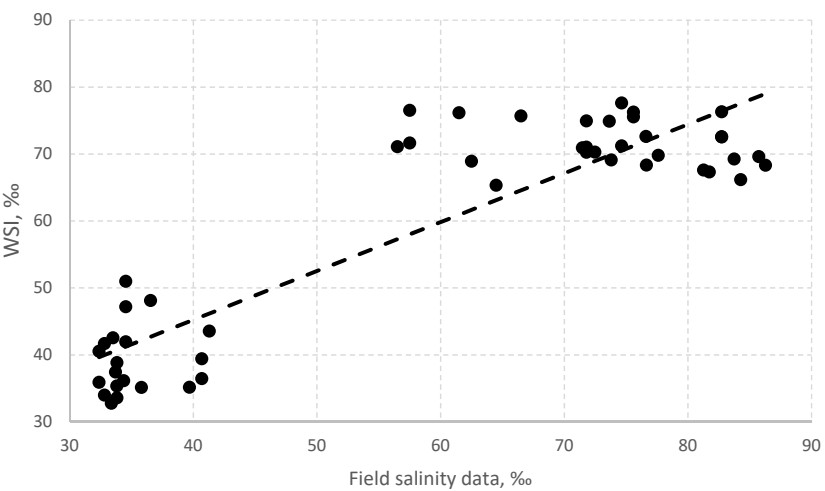

**Figure 7.** A good standard of salinity level prediction can be observed.

The calculated anomaly levels (Figure 6) show almost the same level of data distribution. It should also be noted that contrary to calculations and comparison of model accuracy, the final results appeared to be better precisely with the linear regression model using only the B11 channel. On the other hand, we should not forget about the peculiarities of obtaining raster data from field observations. Since we performed interpolation using the IDW method to obtain them, the data were smoothed over the entire water surface from the sampling points. Hence, it follows that the higher visual accuracy of the linear regression model of the salinity level in the Eastern Sivash is explained by the greater smoothing of the obtained data [29,30].

According to the results, the most accurate method for predicting water salinity is linear regression using the B11 channel. Anomalies between modeling results and interpolated field data were different only in case of linear regression, and other models showed practically the same result (Figures 8 and 9).

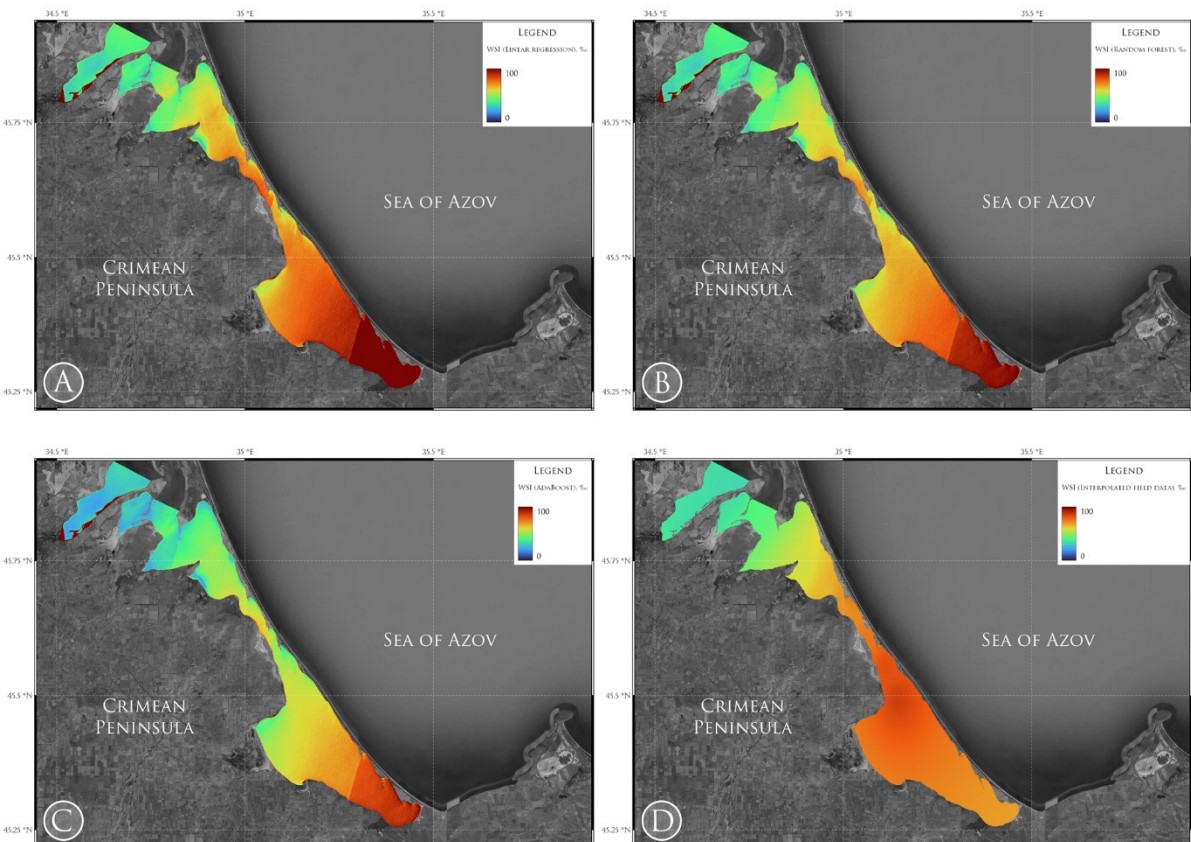

**Figure 8.** Estimated salinity of East Sivash on 23 July 2019 using B11 band with linear regression (**A**), random forest (**B**), AdaBoost (**C**) and field data (**D**).

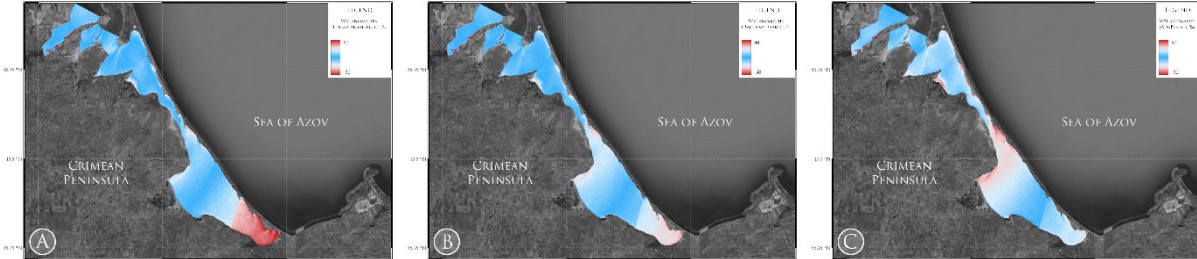

**Figure 9.** Anomalies of water salinity values using B11 band with linear regression (**A**), random forest (**B**) and AdaBoost (**C**) over field data.

Furthermore, we should pay attention to the ultra-high salinity values shown in some coastal areas, bays, etc. In our opinion, the main cause is the fact that at the time of the remote sensing of this area, either the water level there was significantly low or these territories were open soils, as a result of which the data obtained here are significantly overestimated. Hence, it follows that during the long-term monitoring of the salinity (or other environmental parameters) of the Eastern Sivash using remote sensing or other water bodies with similar environmental properties, it is necessary either to exclude ultrahigh anomalous results (by analyzing outliers during post-processing stage) or to form a mask of a water body by extracting water surfaces using, for example, water indices (NDWI, mNDWI, etc.), although with a large amount of data this can significantly increase the simulation time. Therefore, the choice of the correct option should be determined by the researcher, based on the goals, objectives, and technical capabilities of the research.

## 4. Conclusions

In this study, we analyzed the accuracy of the estimation of salinity levels in the eastern part of the hypersaline Sivash Bay with remote sensing data using the machine learning approach. Our results allow us to conclude that our models show an acceptable level of accuracy, with a total error about 10–20%. We also showed that for the long-term monitoring of the salinity level in the eastern part of Sivash Bay, both for fishery and hydrological purposes, it is advisable to use a model based on linear regression or random forest. We also found that with an increase in salinity, the absorbing light shifts from the ultraviolet part of the spectrum to the infrared and short-wave infrared parts, which makes it possible to produce continuous monitoring of hypersaline water bodies using remote sensing data.

**Author Contributions:** Conceptualization, E.Z. and S.C.; methodology, S.C.; software, R.B., V.K. and E.K.; validation, E.K., E.Z. and D.K.; formal analysis, S.C. and D.K.; investigation, E.Z.; resources, R.B., D.K. and V.K.; data curation, S.C.; visualization, V.K.; funding, E.Z. and S.C.; supervision, D.K. and S.C.; project administration, S.C. All authors have read and agreed to the published version of the manuscript.

**Funding:** This some research as part of sub-program of the Azov-Black Sea Branch of the FSBSI "VNIRO" ("AzNIIRKH") "Multidisciplinary study of the aquatic biological resources in the Sea of Azov and Black Sea aimed at conservation of aquatic biological resources and their habitat, which includes state monitoring and determination of total allowable catches and recommended yield of aquatic biological resources in 2020", sub-clause 2.1.1.6.1. The research is partially funded by the Ministry of Science and Higher Education of the Russian Federation as part of the World-class Research Center program: Advanced Digital Technologies (contract No. 075-15-2020-903 dated 16 November 2020).

**Conflicts of Interest:** The authors declare no conflict of interest.

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
