# Peer review of "Surface Water Salinity Evaluation and Identification for Using Remote Sensing Data and Machine Learning Approach"

_jmse, doi:10.3390/jmse10020257_

Round 1

Reviewer 1 Report

A general comment for the whole paper: the authors should avoid using we, us , our .

Title: Needs to be changed to be: Surface Water salinity evaluation using remote sensing data and machine learning approach

Abstract: most of the  sentences need to be re-written so as not to include words like: we, us, we've used, .... etc.  Keywords: to be arranged alphabetically and avoid redundancy 1.  Introduction: References are not cited correctly (for example in L.. 31 there is reference [3] where are references [1 and 2]??. Most of the sentences are missing references and the source obtained. Reference style is not following the journal style. The authors sometimes use numerical style and sometimes use alphabetical. The authors should follow the guide for authors for the references style. Numbering of the headings need to be changed to be as follows: 1. Introduction 2. Water salinity and its absorption properties 3. Materials and Methods 3.1. Remote Sensing of the different environmental variables The last paragraph (L..111-113) to be moved to the end of the introduction section. 3.2. Deep Learning ... 3.3 Research area 3.4 Field data ... 3.5. Remote Sensing Data In L. 181: The main goal of Sentinel-2 instead of: Their main goal is ... Authors need to combine sentences related to each other in a single paragraph. 3.6. Satellite Data... 3.7. Validation All symbols of equations have to be defined clearly in L. 224: where is yb in equation 3? 3.8. Factors selection Table 2": is missing its reference, Negligible not Negligebel 4. Results and Discussion SAVI, VSSI, NDSI and NDVI have to mentioned full name at their first mention.  Funding: very long, needs to be summarized to be more concise References: please refer to guide for authors and revise carefully Language needs to be revised thoroughly

Author Response

Dear Reviewer
Thank you for your comments. We changed title of the manuscript to “Surface Water salinity evaluation using remote sensing data and machine learning approach”. Also, we re-wrote abstract to avoid pronouns and rearranged keywords to alphabetical order. We correct reference style and paragraph numbering.

I hope that this version will fully answer all comments.
Thank you.

Reviewer 2 Report

This regional study mapped salinity level using high remote sensing data, and the method of regression between field samples and remote sensing spectral is commonly used. I have two suggestions below.

1) The dates of field samples and remote sensing imagery are not the same, would this influence the accuracy of regression models? As we know, the tides and wind waves are varying very fast with a short-time period (e.g. hours). 

I suggest trying to use part of samples collected at the same time, or at least at the same day to test the accuracy of developed algorithms. Discuss these potential errors with other studies. Also show the tidal and weather conditions of field samples if it is available.  

Possible references,

Dinnat, E. P., Le Vine, D. M., Boutin, J., Meissner, T., & Lagerloef, G. (2019). Remote sensing of sea surface salinity: Comparison of satellite and in situ observations and impact of retrieval parameters. Remote Sensing11(7), 750.

Zhang, X., Fichot, C. G., Baracco, C., Guo, R., Neugebauer, S., Bengtsson, Z., ... & Fagherazzi, S. (2020). Determining the drivers of suspended sediment dynamics in tidal marsh-influenced estuaries using high-resolution ocean color remote sensing. Remote Sensing of Environment240, 111682.

2) Clarify the expressions, like ' due to some limitations ' in the abstract.

Uniform the decimal numbers.

Author Response

Dear Reviewer
Thank you for your comments. During this research we carefully select time of the samples collecting to standard schedule of the Sentinel-2 data ingestion (regularly 3 days). In 2019 there was one shift from the standard expedition scheme (in June) that was conditioned by other research needs. Also, we specified limitations in abstract.

I hope that this version will fully answer all comments.
Thank you.

Reviewer 3 Report

Dear authors,

The effort to model the salinity of the water using optical remote sensing is interesting and would bring a significant contribution to the field.

Before this paper could be considered for publication, there are a few concerns that need to be addressed.

1. Although the hypothesis is sound and valid, the results acquired in table 3 need more evidence. I suggest to include the scatter plot of the model. In the case of a high correlation between the predictor and in-situ salinity, the machine learning option seems to be out of need. These preliminary results are critical in this study; to provide justification on the fundamental relationship between the reflectance and in-situ salinity.

2. Establishing the multiple regression model of WSI surely would get you a good result as you increase the predictors and degree of freedom; but if one variable is sufficient, you don't have to overdo the modeling by incorporating all. The simpler model would be sufficient and preferred in operational applications.

3. My suggestion is to separate the datasets into two, one is for modeling and one is assessment. I know the number of in-situ samples is limited, but the option for 10 points for modeling and the rest for assessments can be accepted.

4. In the case of the machine learning operation (MLO), it is better to run a comparative analysis; assessing the impact of the MLO in the study. At present, it was loosely highlighted and shown in the manuscript.

I wish that my comment would be helpful in improving the quality of this research.

Thank you.

Author Response

Dear Reviewer
Thank you for your comments. As you suggest, we added scatterplots with comparison of collected samples and modelling results. Also, we changed multiple linear regression on AdaBoost and Random Forest. We separate dataset into training and testing parts to visualize comparison between different models.

I hope that this version will fully answer all comments.
Thank you.

Round 2

Reviewer 1 Report

1. The title is not changed in the revised version as nemntioned by the authors in their response letter. The title need to be changed to:  Surface Water salinity evaluation using remote sensing data and machine learning approach".

2. The language needs to be thoroughly revised by a native language holder because there are lots of mistakes especially after removing the pronouns from the whole paper.

3. Please follow my comments in the attached file

Author Response

Dear Reviewer

Thank you for your comments. We respond to your comments.

"1. The title is not changed in the revised version as nemntioned by the authors in their response letter. The title need to be changed to:  Surface Water salinity evaluation using remote sensing data and machine learning approach"'  

Based on your comment and the comment of other reviewers, we have changed the title of the article.

New Title: Surface Water salinity evaluation and identification for using remote sensing data and machine learning approach

'2. The language needs to be thoroughly revised by a native language holder because there are lots of mistakes especially after removing the pronouns from the whole paper.' 

We've made corrections. Thank you for the file with moments where mistakes were made and we additionally made corrections.

'3. Please follow my comments in the attached file'

Based on your comments, we have corrected the errors you pointed out.
All drawings and formulas were checked again.